# Served Portion Sizes Affect Later Food Intake Through Social Consumption Norms

**DOI:** 10.3390/nu11122845

**Published:** 2019-11-20

**Authors:** Sanne Raghoebar, Ashleigh Haynes, Eric Robinson, Ellen Van Kleef, Emely De Vet

**Affiliations:** 1Consumption and Healthy Lifestyles Group, Wageningen University and Research, Hollandseweg 1, 6706 KN Wageningen, The Netherlands; emely.devet@wur.nl; 2Centre for Behavioural Research in Cancer, Cancer Council Victoria, 615 St Kilda Road, Melbourne, VIC 3004, Australia; Ashleigh.haynes@cancervic.org.au; 3Institute of Psychology, Health, and Society, University of Liverpool, Bedford Street South, Liverpool L69 7ZA, UK; eric.robinson@liv.ac.uk; 4Marketing and Consumer Behavior Group, Wageningen University and Research, Hollandseweg 1, 6706 KN Wageningen, The Netherlands; ellen.vankleef@wur.nl

**Keywords:** portion size, food environment, food intake, social norms, personal norms, portion size normality

## Abstract

Portion sizes of commercially available foods have increased, and there is evidence that exposure to portion sizes recalibrates what is perceived as ‘normal’ and subsequently, how much food is selected and consumed. The present study aims to explore the role of social (descriptive and injunctive) and personal portion size norms in this effect. Across two experiments, participants were either visually exposed to (Study 1, *N* = 329) or actually served (Study 2, *N* = 132) a smaller or larger than normal food portion. After 24 h, participants reported their intended consumption (Study 1) or served themselves and consumed (Study 2) a portion of that food and reported perceived portion size norms. In Study 1, visual exposure to portion size did not significantly affect intended consumption and perceived portion size norms. In Study 2, participants consumed a smaller portion of food when they were served a smaller rather than a larger portion the previous day, which was mediated by perceived descriptive and injunctive social (but not personal) portion size norms. Results suggest that being served (but not mere visual exposure to) smaller (relative to larger) portions changes perceived social norms about portion size and this may reduce future consumption of that food.

## 1. Introduction

Portion sizes of commercially available foods have dramatically increased over recent decades [1,2], and it has been consistently demonstrated that people eat more from larger than from smaller food portions [3,4]. Large food portion sizes have therefore been identified as a possible contributor to the obesity epidemic [5,6]. Recent insights show that portion sizes served in a given eating occasion not only affect immediate consumption, but also affect subsequent portion selection and consumption at later eating occasions [7,8]. Particularly, it has been shown that when served a smaller portion, people select and consume a smaller portion of food in the future compared to when they are served a larger portion. Thus, changes to portion sizes in the environment have potential downstream consequences beyond a single eating occasion [7,8]. However, relatively little is known about the mechanism responsible for this effect [9].

Previous findings suggest that exposure to and consumption of smaller portion sizes may recalibrate perceptions of portion size, making smaller portions more ‘normal’. In one study, mere visual exposure to images of small (versus large) portions decreased subsequent perceptions of what constituted a ‘normal’ sized portion, and this resulted in participants selecting a smaller ideal portion of that food immediately afterwards, but this effect did not translate to actual food selection [10]. Furthermore, in a series of studies, when participants were served a small (versus large) portion of a lunch meal they consumed significantly less one day later, and chose a smaller ideal portion of that same meal one week later [7,8]. It was demonstrated that this effect partly occurs because being served a smaller (versus larger) portion size decreases peoples’ perceptions of what constitutes a ‘normal’ portion size [7]. Comparable results on food intake were found in an experiment manipulating visual exposure to physically present portion sizes of snack foods. In this study, visual exposure to portion sizes affected perceptions of portion size normality 24 h later, although perceptions of portion size normality did not formally mediate the effect of visual exposure on subsequent consumption of the same snack food [8]. In this previous work [7,8], perceptions of portion size normality were measured by asking participants to indicate what they thought was a ‘normal’ portion size of food to eat in a given situation, but it remains unclear what these portion size normality judgments are based on.

Previous research has demonstrated that consumers possess divergent social and personal norms for portion size. A perceived ‘social norm’ represents what consumers believe *other people* consider to be a normal and/or appropriate amount to eat. A perceived ‘personal norm’ represents the amount of food consumers consider to be a normal and/or appropriate amount for *themselves* to eat [11,12]. The construction of a ‘personal norm’ is a dynamic process that is influenced by the external environment [11], and a served portion size may affect future consumption by affecting one’s ‘personal’ norm. There are two conceptually and motivationally distinct types of social norms that may be affected by exposure to different portion sizes. First, consumers may believe a served portion size is based on what other people consume (a ‘descriptive’ social norm) [13]. This is consistent with evidence that social norms about food consumption are inferred from physical aspects of food environments. For example, previous experiments in laboratory and real-world settings have shown that consumers who were presented with a bowl of snacks surrounded by empty snack wrappers consumed more than those who were presented the same snack bowl but without empty wrappers [14,15]. The wrappers may have communicated that others had previously consumed the snacks in the same situation, therefore exemplifying a descriptive social norm communicated by the eating environment. Second, an ‘injunctive’ social norm is what one perceives *ought* to be done [13]. Consumers are likely to assume that a served portion size was not chosen at random by the person serving it, but that there was some reasonable rationale behind providing that amount of food [16]. They may therefore infer that a portion size served to them represents what others think is the appropriate amount for them to eat (an injunctive social norm), and this could affect later portion size selection and consumption.

Rather than via a perceived norm, an alternative explanation is that portion size communicates how much of that food one needs to consume in order to feel satisfied. This ‘expected satiety’ belief may be learned from the post-ingestive consequences of having consumed a given portion (the feeling of satisfaction and the avoidance of hunger after eating), or merely inferred from the amount presented (e.g., ‘this must be enough to keep me satisfied if someone has decided to serve this amount’). Higher expected satiety associated with a given food is associated with the selection of smaller portion sizes and reduced consumption of that food [8,17], and may therefore play a role in the effect of portion size exposure on subsequent behavior.

In the present research, we examined whether visual exposure to (Study 1) and being served (Study 2) smaller versus larger portion sizes would affect later portion size selection for a hypothetical meal (Study 1) and consumption of an actual meal (Study 2). We hypothesized that this effect would be explained by (1) the general perception of what constitutes a normal sized portion, as shown in previous research [7,8], and (2) more specific perceptions of social (both descriptive and injunctive) and personal norms about what is a normal amount to eat in that situation. Particularly, it was expected that visual and actual exposure to smaller (versus larger) food portions would result in a lower intended and actual consumption of that food 24 h later and, accompanying perceptions of a normal-sized portion, perceptions of social (both descriptive and injunctive) and personal norms. We also tested an alternative explanation; that exposure and consumption of smaller portion sizes may affect later portion selection and consumption by affecting expected satiety.

## 2. Study 1

### 2.1. Methods

#### 2.1.1. Design

Study 1 was a two-session online experiment run on two consecutive days with a three-condition between-subjects design. In the first session (initial exposure phase), participants were visually exposed to images of either a relatively small portion of lasagna (smaller portion size condition), a relatively large portion of lasagna (larger portion size condition), or non-food objects (control condition, as in Robinson et al. [10]), to which they were randomly allocated. During the next session on the following day (measurement phase), participants indicated their preferred portion size for a hypothetical lunch meal and their general perceptions of portion size normality. Furthermore, participants indicated their more specific perceptions of descriptive social norms, injunctive social norms and personal norms for portion sizes, in order to examine what these general portion size normality judgments are based on. Additionally, participants indicated their expected satiety regarding portion sizes, as an alternative explanation. A control condition (precluding exposure to portion sizes) was included in the design in order to identify the direction of potential effects. Compared to a control condition, it was expected that visual exposure to smaller (larger) food portions would result in later selection of a smaller (larger) portion size. The same evaluations were made for a different food (spaghetti), to test whether visual exposure effects transfer to an incongruent food [10]. The study was approved by the University of Liverpool research ethics committee (reference number: 3985). The hypotheses, methodology, and main analyses strategy were preregistered on the Open Science Framework (https://osf.io/uznyr) before data collection commenced.

#### 2.1.2. Participants and Sample Size

UK adults (aged 18+ years) were recruited via the online survey platform Prolific Academic. We aimed to recruit 330 participants (approximately 110 participants/condition), which would provide adequate power based on a Monte Carlo power analyses for indirect effects using an online application [18]. Details of the power analyses are provided in the Appendix A.

#### 2.1.3. Portion Size Stimuli

Beef lasagna and spaghetti Bolognese were used as food stimuli (Tesco supermarket). Selection of the portion sizes for the exposure manipulation was informed by the results of an online pilot study conducted with 20 University of Liverpool employees (65% female, M age = 28.65 years, SD = 6.29) (see Appendix A for pilot study results). A relatively small portion (60% of reference portion: 180 g cooked lasagna, 341 kcal) and a relatively large portion (180% of reference portion: 540 g cooked lasagna, 1024 kcal) of lasagna, which were perceived to be beyond the boundaries of a normal portion by the majority of participants, were selected as stimuli for the initial exposure phase (see Figure 1). Portion sizes that were outside of the range perceived as ‘normal’ were selected because this study aims to examine whether portion size norms adjust to these smaller (versus larger) portion sizes, initially perceived ‘not normal’, as one could argue that the range of portion sizes initially classed as being ‘normal’ in size has shifted upwards in recent decades.

#### 2.1.4. Measures

##### Hypothetical Portion Size Selection and Proposed Mediators

Based on the results of the pilot study, an array of nine different portion sizes per food type (lasagna, spaghetti) was selected to create a response scale for the portion selection and mediator items (Figure 1). The response scales ranged from what was perceived by a majority of participants in the pilot study as ‘not normal’ at the scale minimum to a portion that was perceived as a ‘not normal’ amount to eat at the scale maximum. For each outcome and mediator item (e.g., ‘how much of this (lasagna/spaghetti) would you choose to eat for lunch?’), participants responded by selecting one of the nine portion size images that were presented concurrently on screen in ascending order. The specific outcome and mediator items are listed in Table 1. Items were averaged into a single score for variables with multiple items (perceptions of descriptive social norms, injunctive social norms, personal norms and expected satiety). For descriptive and injunctive social norm items, the age and sex of the participants were included in the question text.

#### 2.1.5. Procedure

Participants were invited to participate in a study about consumer perceptions and preferences (cover story). They provided informed consent and were instructed to fill in both questionnaires on a desktop or laptop computer at approximately the same time on consecutive days. The questionnaires were programmed in Qualtrics and were only made active online between 10 am and 2 pm (around a typical lunchtime). During the initial exposure phase, participants viewed an image of a portion of lasagna (either a relatively small or a large portion size) or an image of a non-food object (a printer), depending on their assigned condition. They were instructed to either imagine eating the displayed food during lunch (portion size conditions) or to imagine using the non-food object (control condition) and to write at least five sentences about their imagined experience. Participants then rated the images on 14 dimensions unrelated to portion sizes using 7-point Likert scales (evenly randomized across different pages), and reported their age and sex. The dimensions were either linked to the experience of eating lasagna (smaller and larger portion size condition—e.g., ‘as you eat the lasagna, how (colorful/fresh/crispy) is it’) or to the experience of using the printer (control condition—e.g., ‘as you use the printer, how (colorful/unique/futuristic) is it’).

During the next session (measurement phase), participants again reported their age and sex and then completed the hypothetical portion size selection item, and proposed mediator items (norms and expected satiety, assessed on separate pages in an evenly randomized order). The same measures of hypothetical portion selection, norms, and expected satiety were taken for spaghetti (incongruent food), measured using an array of nine images of spaghetti (ranging from 40% to 200% of the manufacturer’s recommended serving, with a 20% difference between portions, see Appendix A). The order in which participants completed the lasagna and spaghetti questions was (evenly) randomized. Participants then reported their hunger level at the start of the study, their liking for lasagna and spaghetti, demographic information (weight, height, and ethnicity), what they thought was the study aim, and whether they had any allergies or intolerances (see Appendix A). Thereafter, participants completed a funneled manipulation check to assess their recall of the image (including portion size) to which they were exposed (see Appendix A), and were then debriefed and reimbursed.

#### 2.1.6. Planned Statistical Analyses

Data were analyzed using IBM SPSS Statistics 24 (IBM Corp., Armonk, NY, USA). We planned *a priori* to exclude participants from analyses who indicated any allergies, intolerances, or dietary requirements for the foods used in the study, as well as participants who were aware of the study aims.

Separate univariate ANOVAs were performed to examine whether (1) hypothetical portion size selection, (2) perceived portion size normality, (3) perceptions of descriptive social norms, (4) perceptions of injunctive social norms, (5) personal norms, or (6) expected satiety for lasagna varied between the experimental conditions. Bonferroni-corrected pairwise comparisons were examined to probe significant effects. As the variables (1–6) were not normally distributed, data were log-transformed with a natural logarithm before testing and inclusion in further analyses.

The PROCESS macro for SPSS (model 4) [19] was used to investigate whether the effect of condition on later portion size selection could be explained by (a) general perceptions of portion size normality (testing for single mediation) and/or (b) more specific perceptions of descriptive social norms, injunctive social norms, personal norms and expected satiety (testing for multiple mediation). The percentile bootstrapping method was applied, producing 95% confidence intervals for the indirect effect, derived from 5000 bootstrap resamples. Proposed mediators were only included in mediation analyses when the conditions for mediation were met—that is, when the two components of the indirect effect of a proposed mediator were both significant in separate linear regression analyses [20].

To detect potential transfer effects, all analyses were repeated using portion size selection, norms, and expected satiety for spaghetti (incongruent food).

Pearson correlation coefficients (Spearman’s correlations for sex) between potential covariates (age, sex, BMI, exposure duration, hunger, liking, and ethnicity) and the outcome variable or mediator variables were examined. Sensitivity analyses were run, which repeated the main analyses, controlling for covariates that were significantly correlated.

### 2.2. Results

#### 2.2.1. Participant Characteristics

In total, 338 participants completed both phases on two consecutive days. Nine participants were excluded from analyses in line with *a priori* exclusion criteria (see Appendix A). The analytic sample consisted of 329 participants (see Table 2 for participant characteristics per condition).

#### 2.2.2. Funneled Manipulation Check

Participants (324/329, 98.5%) tended to correctly identify the lasagna or non-food object they were exposed to during the first part of the study. Two participants in the smaller portion size condition, two participants in the larger portion size condition, and one participant in the control condition were not able to correctly identify the lasagna or non-food object. Likelihood ratio chi-square analyses showed no significant difference in correct identification between the visual exposure conditions, Λ(2) = 0.37, *p* = 0.83. Among participants who correctly selected lasagna in the first manipulation check question, Pearson chi-square analyses showed that a higher proportion of participants in the smaller portion size condition (67/107, 62.6%) than the larger portion size condition (33/117, 28.2%) were able to correctly identify the portion size they were exposed to, *X*^2^(1) = 27.30, *p* < 0.001. Results of a Welch’s *t*-test indicated that participants in the smaller portion size condition (M = 3.18, SD = 1.71, *N* = 105) remembered being exposed to a significantly smaller portion than participants in the larger portion size condition (M = 5.30, SD = 2.09, *N* = 115) (*t*(215.57) = −8.27, *p* < 0.001, *d* = 1.10).

#### 2.2.3. Hypothetical Portion Size Selection, Norms, and Expected Satiety

There were no significant differences between conditions in hypothetical portion size selection, perceived portion size normality, perceptions of descriptive social norms, perceptions of injunctive social norms, personal norms, or expected satiety regarding portions of lasagna during the second session, although all scores were in the predicted direction (Table 3). A similar pattern of results was observed for spaghetti (Table 3). There was a marginally significant effect of condition on perceptions of injunctive social norms for spaghetti, although there were no significant differences between the smaller portion size condition and larger portion size condition (*p* = 0.18), between the smaller portion size condition and control condition (*p* = 0.16), or between the larger portion size condition and control condition (*p* = 0.99). Mediation analyses were not conducted for either lasagna or spaghetti, as the conditions for mediation were not met (see Appendix A). The results of the sensitivity analyses are reported in Appendix A.

#### 2.2.4. Unregistered Exploratory Analyses

Participants in the larger portion size condition were less likely to identify the portion size they were exposed to than participants in the smaller portion size condition. Poor recall of the manipulation may have been responsible for the pattern of results from primary analyses. Therefore, primary analyses were repeated on the subsample of participants who correctly identified the portion size they were exposed to (*n* = 204). As in the primary pre-registered analyses, exposure condition did not influence later portion size selection for lasagna. However, perceived portion size normality, descriptive social norms, injunctive social norms, personal norms, and expected satiety regarding portions of lasagna were significantly different between experimental conditions (see Appendix A). Bonferroni-corrected pairwise comparisons showed a significant difference between the smaller portion size and larger portion size conditions regarding perceptions of portion size normality (*p* = 0.001), descriptive social norms (*p* = 0.01), injunctive social norms (*p* = 0.002), personal norms (*p* = 0.03), and expected satiety (*p* = 0.02), in that reported norms and expected satiety were relatively smaller in the smaller portion size condition. Differences between the larger portion size condition and the control condition were in the expected direction but did not reach criteria for statistical significance, as follows: Perceptions of portion size normality (*p* = 0.053), injunctive social norms (*p* = 0.08), and personal norms (*p* = 0.055). No significant differences were observed between the larger portion size condition and the control condition regarding perceptions of descriptive social norms or expected satiety (both *p* = 0.12). No significant differences between the smaller portion size condition and control condition were observed for any of the proposed mediators (all *p*-values > 0.18).

### 2.3. Conclusions

Exposure to images of smaller (relative to larger) portion sizes did not affect hypothetical portion size selection or perceived normality 24 h later. However, in unplanned exploratory analyses on the subset of participants who correctly recalled the manipulation, visual exposure to smaller (compared to larger) portions of lasagna decreased perceptions of a normal-sized portion, descriptive social norms, injunctive social norms, personal norms and expected satiety regarding portions of lasagna 24 h later, although there was no significant effect of visual exposure to portion sizes on later portion selection. This pattern of results may suggest that the online portion size exposure manipulation was not salient enough, or that it may be hard to make inferences about food portion sizes from visual stimuli. Although previous studies have demonstrated immediate effects of mere visual exposure in an online setting on subsequent perceived portion size norms and intended consumption [10], this is the first study to examine effects over a longer time period (24 h later). A stronger manipulation (via exposure to physically present portion sizes or actual consumption of different portion sizes) may be required to investigate the precise normative mechanisms underlying this phenomenon. Therefore, Study 2 was conducted in a laboratory setting in which participants were served and then consumed either a smaller or larger portion of lasagna, before returning 24 h later to serve themselves a portion of that same food and to report their perceived portion size norms. As Study 1 provided no evidence of transfer to an incongruent food, consumption and normative evaluations were assessed only for lasagna.

## 3. Study 2

### 3.1. Methods

#### 3.1.1. Design

Study 2 was a two-session laboratory-based experiment run on two consecutive days with a two-condition between-subjects design. Participants were served and then consumed a relatively small portion of lasagna (smaller portion size condition) or a relatively large portion of lasagna (larger portion size condition) in the first session (manipulation phase). During the session on the next day (measurement phase), participants were presented with a family-sized lasagna (see Figure 2c) and were instructed to serve themselves and consume whatever they want to eat. Thereafter, participants completed the same measures of perceived portion size normality, descriptive social norms, injunctive social norms, personal norms, and expected satiety regarding portion sizes of lasagna as in Study 1. The research ethics committee of Wageningen University and Research approved the study (CoC number 09215846). The study was preregistered on the Open Science Framework (https://osf.io/7rqa4/) before data collection commenced.

#### 3.1.2. Participants and Sample Size

Sex was significantly correlated with all outcome and mediator measures in Study 1 (*p* < 0.01). Furthermore, previous research found that women are more likely to follow social norms when eating, compared to men [21], and it has been shown that there are differences in portion size preferences between men and women [12,22]. Therefore, only females were recruited in Study 2 to minimize heterogeneity in food consumption, as sex differences in portion size preference and evaluations might affect the portion size exposure manipulation. Dutch female students and employees of Wageningen University and Research were included in the study. To be eligible to participate, participants were required to be willing to eat beef lasagna, and participants who followed a diet; did not consume beef lasagna because of allergies, intolerances, or dietary specific requirements (e.g., vegetarian); or participants who were included in our pilot study were ineligible to participate.

As in Study 1, the sample size was determined based on a Monte Carlo power analyses for indirect effects using an online application [18]. We aimed to recruit 140 participants (approximately 70 participants/condition). Details of the power analyses are provided in the Appendix A.

#### 3.1.3. Portion Size Stimuli

Beef lasagna was used as food stimuli (Aldi supermarket). Selection of the smaller and larger portion sizes (Figure 2) was informed by the results of a pilot study conducted in a sample of Dutch female participants, including students (83%) and employees of Wageningen University and Research (M age = 23.80 years, SD = 4.12, *n* = 51; M BMI = 21.65, SD = 2.04, *n* = 51). As in Study 1, a relatively small portion (60% of reference portion) and a relatively large portion (180% of reference portion) of lasagna, which were perceived to be beyond the boundaries of a normal portion by the majority of participants, were selected for the manipulation phase (see Appendix A for pilot study results).

#### 3.1.4. Measures

##### Portion Size Selection and Consumption and Proposed Mediators

Consumption of lasagna in session two was calculated in grams by subtracting the post-lunch weight of the leftover family-sized lasagna from the pre-lunch weight. Portion size selection (i.e., the amount of food that participants removed from the serving tray) was measured by subtracting the post-lunch weight of the family-sized lasagna from the pre-lunch weight. Salad selection and consumption was measured in the same way.

Similar items as in Study 1 were used to measure portion size normality, descriptive social norms (Cronbach’s α = 0.84), injunctive social norms (Cronbach’s α = 0.87), personal norms (Cronbach’s α = 0.91), and expected satiety (Cronbach’s α = 0.87) in Study 2, with a mean score calculated for variables consisting of more than one item. See Table 1 for the specific mediator items. The only difference from Study 1 was that items measuring perceptions of descriptive social norms and injunctive social norms referred to a situation-specific referent group, ‘female participants of this research study’, rather than personalizing these items to their age and sex (e.g., ‘how much of this lasagna do you believe other female participants of this research would choose to eat for lunch?’). Given that the results of the pilot study for Study 2 were similar to Study 1 (see Appendix A), the same 9-portion response scale for the mediator items was used in Study 2.

#### 3.1.5. Procedure

Participants were invited to participate in a study about the influence of the way a lunch meal is served on their mood (cover story). Participants attended two sessions on consecutive weekdays between 12 pm and 2 pm (a typical Dutch lunchtime) and the experiment was conducted in a sensory laboratory which consisted of five cubicles. Eligible participants were asked to abstain from eating for at least two hours before each session. They provided informed consent for both sessions at the start of session 1. To bolster the cover story, participants completed mood questionnaires before and after eating lunch, including items measuring appetite and one item asking participants to report how long since they last ate (see Appendix A). Participants were instructed to press a button when they had completed the questionnaire, at which point the researcher returned with the lunch. In both conditions (small, large) the lunch was served on a standard white dinner plate (Ø 28.5 cm), consisting of lasagna and a 10 g side salad (lettuce leaves), served along with a glass of water, cutlery, and a napkin. Participants were served a portion of lasagna corresponding to the condition to which they were randomly assigned according to a predetermined computerized random sequence of conditions. All participants were told that they could consume the entire meal if they wanted to and were instructed to press a button to alert the researcher when they had finished eating. Participants then completed the same mood and appetite ratings as before lunch. On the next page of the online questionnaire, participants reported how much they liked the lasagna (see Appendix A), which was embedded alongside two filler items about the palatability of the lasagna (e.g., “how much did you like the smell of the lasagna”) measured on 9-point scales ((1 (not at all) to 9 (extremely)). Items were presented in an evenly randomized order.

During the second session (scheduled for approximately the same time on the day following session one), participants first completed the same mood, appetite, and time since last eaten questionnaire items as in session one. The researcher then returned with an empty standard white dinner plate (Ø 28.5 cm), cutlery, a napkin, and a glass of water and a tray consisting of a family-sized lasagna in an aluminum container, a full bowl of salad (lettuce leaves (30 g)), and serving utensils (see Figure 2c). The researcher informed participants that they could serve themselves whatever they wanted to eat and requested they place the tray containing the unserved food behind a hatch after serving themselves. Participants pressed the button when they finished eating, at which point the researcher returned to retrieve all remaining food, cutlery, and water, and unobtrusively measured the amount of lasagna that participants selected and consumed. Then, participants completed the same mood, appetite, and palatability rating questionnaires as in the previous session, and subsequently reported what they thought was the aim of the study (as in Study 1), and subsequently indicated their portion size norms and expected satiety. Participants’ then completed the manipulation check, frequency of eating lasagna (see Appendix A), and demographic items (weight, height, age, and nationality). Thereafter, participants completed the awareness of monitoring consumption item (see Appendix A), which was embedded alongside four filler items about the participants’ study experience (e.g., “did you feel bored during the study”, measured on a 9-point scale ranging from 1 (not at all) to 9 (very much)). Participants were thanked and reimbursed and after completion of all data collection, participants were debriefed about the true purpose of the study by email.

#### 3.1.6. Planned Statistical Analyses

Data were analyzed using IBM SPSS Statistics 24 (IBM Corp., Armonk, NY, USA). We planned *a priori* to exclude participants from analyses who were aware of the study aims or who did not follow study instructions (e.g., not adhering to abstinence requirements).

The analysis plan was the same as in Study 1, except that separate independent-sample *t* tests replaced ANOVAs for examining effects on consumption, portion size selection, and portion size evaluations. As the proposed mediator variables were not normally distributed, data were log-transformed with a natural logarithm. Additionally, as the effect of portion size served in session one on consumption and portion size evaluations the next day may have been dependent on participants’ recollection of the portion size being served during the first session (as suggested by the results of Study 1), primary analyses were rerun by solely including participants who correctly identified the portion size served during session one in the analysis. We reported whether these sensitivity analyses resulted in findings that differed from the primary analyses. See Appendix A for details of additional secondary and sensitivity analyses.

### 3.2. Results

#### 3.2.1. Participant Characteristics

In total, 140 eligible participants completed both sessions of the experiment. Eight participants were excluded from the analyses in line with *a priori* exclusion criteria (see Appendix A), leaving an analytic sample of 132 participants. See Table 4 for participant characteristics per condition.

#### 3.2.2. Manipulation Check

As in Study 1, Pearson chi-square analyses showed that a higher proportion of participants in the smaller portion size condition (68/68, 100%) were able to identify the portion size they consumed in session one than in the larger portion size condition (24/64, 37.5%), *X*^2^(1) = 60.98, *p* < 0.001. Results of a Welch’s *t*-test indicated that participants in the smaller portion size condition (M = 1.32, SD = 0.47) remembered being served a significantly smaller portion than participants in the larger portion size condition (M = 5.94, SD = 1.83) (*t*(70.82) = −19.54, *p* < 0.001, *d* = 3.51), however the proportions were higher than in Study 1, suggesting a better recall of portions when they are served and consumed than when visually presented online.

#### 3.2.3. Consumption and Portion Size Selection

On average, participants who were served the smaller portion size on day 1 consumed almost the entire serving (179.60 g consumed (SD = 15.94) versus 180 g served), while this was not observed in participants who were served the larger portion size on day 1 (450.51 g consumed (SD = 87.86) versus 540 g served). Results of a Welch’s *t*-test showed that participants in the smaller portion size condition (compared to the larger portion size condition) consumed significantly less lasagna during day 1 (*t*(66.96) = −24.29, *p* < 0.001, *d* = 4.34, *n* = 67 for the smaller portion size condition (because of a missing observation)).

Participants who were served the smaller (as opposed to the larger) portion size during day 1 freely chose to select and consume a significantly smaller portion of that food the next day (see Table 5). Results of secondary consumption analyses (i.e., salad consumption) are reported in the Appendix A.

#### 3.2.4. Portion Size Evaluations

Contrary to our hypothesis, there was no significant difference between smaller and larger portion size conditions on perceived portion size normality, or on personal norms or expected satiety (see Table 5). However, consistent with the hypotheses, participants who were served the smaller (as opposed to the larger) portion size during session one reported significantly smaller descriptive and injunctive social norms regarding portions of that food the next day (see Table 5).

A mediation analyses including only perceptions of descriptive social norms and injunctive social norms was performed, as the conditions for mediation were only met for these mediators (see Appendix A). The total indirect effect of portion size condition on later consumption jointly via perceptions of descriptive social norms and injunctive social norms was significant (indirect effect = 18.10, SE = 9.22, 95% CI (2.78, 38.37), proportion of total effect explained by indirect effect = 20.28%). Focusing on specific indirect effects, there was a significant indirect effect of portion size condition on later consumption via perceptions of descriptive social norms (indirect effect = 13.75, SE = 8.37, 95% CI (0.52, 32.99), proportion of total effect explained by indirect effect = 15.41%), but no specific indirect effect via injunctive social norms was observed (indirect effect = 4.35, SE = 6.74, 95% CI (−7.62, 19.68), proportion of total effect explained by indirect effect = 4.87%). The results of the sensitivity analyses are reported in the Appendix A.

#### 3.2.5. Recollection of Portion Sizes

Although the proportion of participants that correctly recalled the portion size served in session one was higher than in Study 1, the majority of participants in the larger portion size exposure condition did not correctly recall the portion size served. Primary analyses were repeated on the subsample of participants in the analyses who correctly identified the portion size served during session one (*n* = 92). The results were all in the predicted direction among this subsample of participants, as follows: Participants in the smaller portion size condition perceived a smaller portion size to be normal; reported a smaller perceived descriptive, injunctive, and personal portion size norm; reported expected satiety with a smaller portion size; and selected and consumed significantly less in the second session than participants in the larger portion size condition (all *p*-values < 0.01, see Appendix A).

### 3.3. Conclusions

Participants who were served a smaller (compared to a larger) portion of lasagna chose to consume a significantly smaller portion of that food the next day. This effect was mediated by differences in the perceived descriptive and injunctive social norms regarding portions of lasagna, but only changes to perceived descriptive norms showed evidence of independent mediation. Neither perceptions of portion size normality, personal norms, nor expected satiety regarding portions of lasagna the next day were significantly different between participants who were served a smaller versus a larger portion of lasagna during session one. In line with Study 1, the pattern of results differed somewhat among the subsample of participants who correctly recalled the portion size they had been served during the first session. In this subsample, being served a smaller (compared to a larger) portion of lasagna resulted in participants believing a normal portion of lasagna to be smaller the next day, and it also decreased their evaluations of personal norms and expected satiety regarding portions of that food the following day. However, as in Study 1, results of these subsample analyses should be interpreted with caution as sample sizes were unequal between conditions.

## 4. Discussion

Across two experiments, we tested whether perceptions of descriptive social norms (beliefs about what others do), injunctive social norms (beliefs about what should be done according to others), and personal norms (beliefs about what should be done according to oneself) underlie the effect of exposure to different food portion sizes on future portion size selection and consumption of the same food. The present findings indicate that participants who were actually served a smaller (versus larger) portion size of food served themselves and consumed less of that food the next day (Study 2), whereas mere visual exposure to a smaller (versus larger) portion size of food did not affect hypothetical portion size selection the next day (Study 1). Consistent with our hypotheses, Study 2 found that the relationship between the portion size exposure condition and later consumption was partially mediated by changes in perceptions of descriptive and injunctive social norms (but not personal norms) for portions of that food, although, contrary to predictions, no significant evidence was found for the role of general perceptions of portion size normality in this relationship.

The results of the present research are consistent with previous findings, demonstrating that *consuming* smaller (versus larger) food portions decreases the amount of food that participants later freely serve themselves and consume [7,8]. Together, this evidence supports the proposition that downsizing commercially available food products could have effects that extend beyond the consumption of the reduced food products, by affecting future portion size preference and consumption [7]. An alternative proposition that has been explored in other research is that visual exposure to smaller food portions via digital media may ‘renormalize’ small portions. A feasibility study of a social media intervention that involved exposing students to images of peers’ snacks in small portion sizes resulted in participants reporting a smaller ideal snack portion size [23]. However, the present findings that mere visual exposure to images of smaller (versus larger) portions was insufficient to significantly decrease future hypothetical consumption of that food reinforces the suggestion that food portions may need to be physically present in one’s environment in order to be able to adjust perceptions of a normal-sized portion and future consumption [8]. Systematic exploration of the necessary conditions for altering perceived portion size normality and future consumption would be valuable to inform effective strategies to reduce overconsumption.

The present work did not replicate earlier findings that exposure to portion sizes affects consumers’ general perceptions of what is a ‘normal-sized’ portion. This was unexpected, as it has been repeatedly shown that exposure to a stimulus can alter people’s perceptions of size normality, demonstrated in relation to food portions [7,8,10], as well as in other domains (e.g., perceptions of ‘normal’ body sizes) [24,25]. A possible explanation for these non-significant findings could be that the more nuanced questions about portion size norms included in the present research (e.g., descriptive and injunctive social norms, personal norms) prompted participants to think more deeply about their beliefs about portion size, dampening the effect of prior exposure to portion sizes on their reported perceived ‘portion size normality’, in general terms. A different interpretation of the non-significant findings might be that the extreme small and large portion sizes included in our experiments were too different from what participants initially perceived as being ‘normal’ in size, as the portion sizes (both small and large) included in the initial exposure phase were selected based on their similar deviance from a normal portion.

Consistent with expectations, current findings suggest that perceptions of descriptive and injunctive social norms jointly underlie the effect of being served (but not visually exposed to) a smaller compared to a larger food portion on consumption of that food 24 h later. To our knowledge, this is the first study empirically showing that served portion sizes can signal normative information about both what others would eat (descriptive social norm) and what others believe is the appropriate amount to eat (injunctive social norm), indirectly affecting future consumption. It should be noted that next to the *combined* indirect effect of descriptive and injunctive norms, there was a *specific* indirect effect of descriptive social norms, suggesting that this factor is relatively more important than perceptions of injunctive social norms. However, when including covariates in the model, only the *total* indirect effect of both descriptive social norms and injunctive social norms remained significant and these norms were highly correlated. Therefore, results should be interpreted with a focus on the overall pattern of the joint indirect effect, which indicates that portion size norms are anchored in social groups [3,11]. Our work supports the notion that social norms can affect eating behavior (e.g., see reviews of Higgs [26], Robinson, Thomas, Aveyard, and Higgs [27], and Stok, de Vet, de Ridder, and de Wit [28]). Furthermore, our research is one of the first studies that provides empirical evidence for the proposition that social norms are embedded in physical elements of food environments, guiding eating behavior accordingly [29].

Contrary to expectations, the indirect effect of portion size condition on consumption via personal norms was not supported, suggesting that exposure to portion sizes does not adjust the amount of food an individual considers to be a normal amount for *themselves* to eat. We also did not find evidence for the alternative explanation that expected satiety underlies the effect of portion size exposure on later consumption. The results of the current study may therefore indicate that portion size norms are derived from the specific eating situation [30,31], rather than reflecting an individualized norm (e.g., this is normal for me to eat or this is enough for me to feel satisfied), as only significant effects for social consumption norms were observed. Specifically, when participants were served a food portion size which was inconsistent with their personal norm in the present research, they may have inferred that this portion size signals what is seen by others in that context as a ‘normal’ or ‘appropriate’ amount to eat. As a result, they conform to these social norms, which might explain the indirect effects demonstrated in this and other studies [7,8]. In other words, serving smaller (larger) portions leads to smaller (higher) food intake, as people think others in that eating context believe that this is the normal/appropriate amount for them to eat, and this effect occurs even if the portions were initially deemed abnormal and irrespective of their own personal standards. This line of reasoning is consistent with Herman and Polivy [31], who argue that portion sizes determined by another person represent a judgment about what one ought to eat in a specific eating situation. Future research should examine whether the observed effects extend to more realistic settings (e.g., at home) in which social norms may be less salient [26]. One could reason that, in a home setting, participants may be more strongly guided by a personal norm (which was not significantly affected by the portion size manipulation in the present research) than a social norm.

In the present research, a large number of participants incorrectly recalled the portion size to which they had previously been exposed. Participants who were exposed to (Study 1) or served (Study 2) larger (versus smaller) portion sizes had a poorer recall of the exposure portion size, which is in line with previous research indicating a general underestimation of especially large portion sizes [32]. The effects of exposure to portion sizes on norm perceptions and expected satiety were potentially stronger (and in the predicted direction) among those participants who were able to correctly identify the portion size to which they were exposed to. This finding may suggest a potential role for making people aware of the portion size as a factor to moderate the relation between portion size exposure and later portion size evaluations, although recent research has failed to demonstrate that training attentive or mindful eating successfully reduces intake [33,34].

Study 1 was conducted via an online survey platform, and therefore may be susceptible to bias due to lack of participant attention. This could potentially explain the observation that fewer participants in Study 1 (correct identification: 44.6%) were able to correctly identify the portion size to which they were exposed than in Study 2 (correct identification: 69.7%). The exclusion of male participants from Study 2 means that it is unclear whether our findings are generalizable to men, and future research should replicate this study including males. Another limitation of this research is that the social and personal norm measures were not validated, although these instruments were developed based on the general norm measure used in Robinson & Kersbergen [7]. Hunger was measured retrospectively in Study 1 and this is a limitation of the current research, although this method has been used previously [10]. Lasagna and spaghetti were selected as stimuli in the present experiments, as they are commonly consumed and are widely available in mainstream supermarkets in both the UK and the Netherlands. However, we did not assess participants’ familiarity with the foods in either study, or their frequency of consumption in Study 1, and we therefore cannot conclude that the foods were equally as well-known across the studies. It is possible that these factors moderate the effect of portion size exposure on future portion size preferences and this could explain the different pattern of results between the studies, but this remains a question for future research. The current research does not permit conclusions about whether exposure to smaller versus larger portion sizes would affect participants’ urge to compensate for a small portion size by eating additional food, but this would be a valuable direction for future research. Although there were no significant effects of visual exposure to portion sizes on normative evaluations of an ‘incongruent’ food in Study 1, future research should test these ‘transfer’ effects in a laboratory-based setting using actual food. Finally, it is unclear whether the effects that portion size had on social norms in the present studies were context specific (i.e., specific to portion size evaluations when eating in laboratory) or whether they would transfer and influence perceived social norms in other contexts (i.e., outside of the laboratory setting).

## 5. Conclusions

Being served (but not being visually exposed to) smaller food portions decreases consumers’ perceptions of social norms regarding both what others would serve themselves (descriptive) and what they believe is an appropriate amount to eat (injunctive), which reduces consumption of that food 24 h later.

## Figures and Tables

**Figure 1 nutrients-11-02845-f001:**
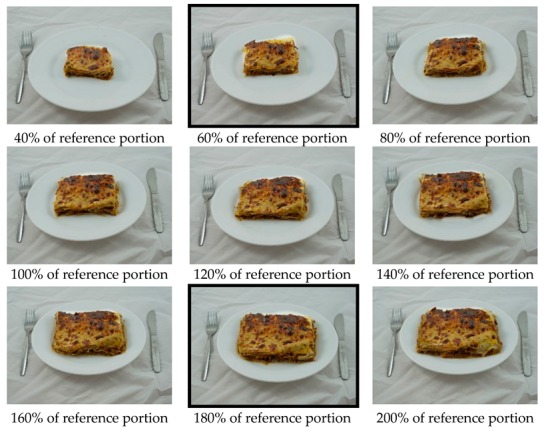
Portion size scale for lasagna ranging from 40% to 200% of the manufacturer’s recommended serving with a 20% difference between portions. The highlighted portions were used as the smaller and larger portion sizes in the first session of Study 1.

**Figure 2 nutrients-11-02845-f002:**
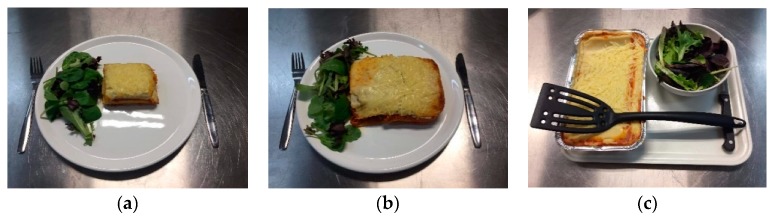
Lasagna served in Study 2: (**a**) Smaller portion size (180 g cooked lasagna, 292 kcal); (**b**) larger portion size (540 g cooked lasagna, 875 kcal); and (**c**) family-sized lasagna (978 g cooked lasagna, 1584 kcal).

**Table 1 nutrients-11-02845-t001:** Items measuring hypothetical portion size selection and proposed mediators.

Measures	Items
Hypothetical portion size selection	1. ‘how much of this (lasagna/spaghetti) would you choose to eat for lunch?’ [7]
Perceptions of general portion size normality	1. ‘how much of this (lasagna/spaghetti) would you say is normal to eat for lunch?’ [7]
Perceptions of descriptive social norms (lasagna: Cronbach’s α = 0.92; spaghetti: Cronbach’s α = 0.94)	1. ‘how much of this (lasagna/spaghetti) do you believe other (sex) aged (age) years would choose to eat for lunch?’
	2. ‘how much of this (lasagna/spaghetti) do you believe most (sex) aged (age) years would choose to eat for lunch?’
Perceptions of injunctive social norms (lasagna: Cronbach’s α = 0.89; spaghetti: Cronbach’s α = 0.90)	1. ‘how much of this (lasagna/spaghetti) do you think other (sex) aged (age) years believe that you are supposed to eat for lunch?’
	2. ‘how much of this (lasagna/spaghetti) do you think other (sex) aged (age) years believe that is the appropriate amount to eat for lunch?’
Personal norms (lasagna: Cronbach’s α = 0.90; spaghetti: Cronbach’s α = 0.91)	1. ‘how much of this (lasagna/spaghetti) do you personally believe is a normal amount for you to eat for lunch?’
	2. ‘how much of this (lasagna/spaghetti) do you personally believe is an appropriate amount for you to eat for lunch?’
Expected satiety (lasagna: Cronbach’s α = 0.90; spaghetti: Cronbach’s α = 0.91)	1. ‘how much of this (lasagna/spaghetti) would you need to eat to feel satisfied?’
	2. ‘how much of this (lasagna/spaghetti) would you need to eat to feel full?’

**Table 2 nutrients-11-02845-t002:** Participant characteristics per condition (*n* = 329).

	Smaller Portion Size Condition (*n* = 107) ^b^	Larger Portion Size Condition (*n* = 117) ^c^	Control Condition (*n* = 105) ^d^
	Mean (SD) or Number (%)	Mean (SD) or Number (%)	Mean (SD) or Number (%)
Age (years)	39.08 (13.35)	37.77 (11.84)	38.33 (10.84)
Sex (female)	79 (74.5%)	73 (62.4%)	61 (58.7%)
BMI (kg/m^2^)	26.93 (6.18)	26.92 (6.24)	25.95 (4.97)
Exposure duration (mm:ss)	04:06 (02:00)	04:57 (04:25)	04:56 (05:21)
Hunger ^a^	4.03 (2.12)	4.82 (2.46)	4.18 (2.24)
Liking ^a^			
Lasagna	7.29 (1.92)	6.76 (2.37)	7.25 (1.93)
Spaghetti	7.08 (1.91)	6.74 (2.39)	7.11 (1.76)
Ethnicity (white)	102 (95.3%)	105 (89.7%)	100 (95.2%)

^a^ Measured on a 9-point scale (range 1–9). ^b^
*n* = 106 for age and sex and *n* = 103 for BMI. ^c^
*n* = 115 for age and *n* = 114 for BMI. ^d^
*n* = 104 for age, sex, and BMI. Reasons for missing values were response inconsistencies between session 1 and 2 for age and sex and implausible responses for BMI (i.e., an unrealistic low or high reported weight or length).

**Table 3 nutrients-11-02845-t003:** Portion size selection and portion size evaluations per condition on day 2 (*n* = 329).

		Smaller Portion Size Condition (*n* = 107)	Larger Portion Size Condition (*n* = 117)	Control Condition (*n* = 105)	Test Statistic	*p*-Value	η_p_^2^
		Mean (SD)	Mean (SD)	Mean (SD)			
**Effect of condition on portion size selection**							
Portion size selection ^a^	Lasagna	3.92 (2.36)	4.07 (2.29)	3.85 (2.10)	*F*(2, 326) = 0.08	0.92	0.001
	Spaghetti	3.74 (1.92)	3.79 (1.74)	3.69 (1.73)	*F*(2, 326) = 0.11	0.90	0.001
**Effect of condition on perceptions of portion size normality**							
Perceptions of portion size normality ^a^	Lasagna	3.57 (1.68)	3.84 (1.63)	3.70 (1.55)	*F*(2, 326) = 0.77	0.46	0.01
	Spaghetti	3.41 (1.34)	3.61 (1.35)	3.50 (1.36)	*F*(2, 326) = 0.68	0.51	0.004
**Effect of condition on perceptions of descriptive norms, injunctive norms, personal norms and expected satiety**							
Perceptions of descriptive norms ^a^	Lasagna	3.74 (1.88)	4.23 (1.91)	3.95 (1.94)	*F*(2, 326) = 1.59	0.21	0.01
	Spaghetti	3.58 (1.60)	3.85 (1.66)	3.91 (1.64)	*F*(2, 326) = 1.14	0.32	0.01
Perceptions of injunctive norms ^a^	Lasagna	3.29 (1.63)	3.65 (1.68)	3.56 (1.75)	*F*(2, 326) = 0.97	0.38	0.01
	Spaghetti	3.14 (1.41)	3.50 (1.52)	3.57 (1.50)	*F*(2, 326) = 2.41	0.09	0.02
Personal norms ^a^	Lasagna	3.49 (1.85)	3.85 (1.77)	3.44 (1.71)	*F*(2, 326) = 2.00	0.14	0.01
	Spaghetti	3.36 (1.53)	3.57 (1.46)	3.40 (1.49)	*F*(2, 326) = 0.80	0.45	0.01
Expected satiety ^a^	Lasagna	4.12 (1.91)	4.35 (1.92)	4.07 (1.91)	*F*(2, 326) = 0.47	0.62	0.003
	Spaghetti	3.84 (1.69)	3.94 (1.56)	3.82 (1.68)	*F*(2, 326) = 0.44	0.65	0.003

^a^ Measured on a 9-point scale (range 1–9). Note: All reported means and standard deviations are untransformed scores for ease of interpretation.

**Table 4 nutrients-11-02845-t004:** Participant characteristics per condition (*n* = 132).

	Smaller Portion Size Condition (*n* = 68)	Larger Portion Size Condition (*n* = 64)
	Mean (SD) or Number (%)	Mean (SD) or Number (%)
Age (y)	20.75 (1.84)	21.11 (2.21)
BMI (kg/m^2^)	21.71 (2.37)	22.04 (2.25)
Baseline hunger (session two) ^a^	6.97 (1.46)	7.13 (1.58)
Liking (session two) ^a^	6.15 (1.45)	6.02 (1.60)
Frequency of eating lasagna ^a^	5.13 (0.98)	5.22 (0.93)
Awareness of monitoring consumption ^a^	7.13 (1.99)	6.69 (2.22)
Nationality (Dutch)	66 (97.1%)	59 (92.2%)

^a^ Measured on a 9-point scale (range 1–9).

**Table 5 nutrients-11-02845-t005:** Portion size selection, consumption and portion size evaluations per condition on day 2 (*n* = 132).

	Smaller Portion Size Condition (*n* = 68)	Larger Portion Size Condition (*n* = 64)	Test Statistic	*p*-Value	*d*
	Mean (SD)	Mean (SD)			
**Effect of condition on consumption**					
Portion size selection (grams)	401.64 (115.25)	505.24 (135.00)	*t*(130) = −4.75	<0.001	0.83
Consumption (grams)	382.57 (104.70)	471.81 (120.91)	*t*(130) = −4.54	<0.001	0.79
**Effect of condition on perceptions of portion size normality**					
Perceptions of portion size normality ^a^	3.04 (1.09)	3.38 (1.43)	*t*(130) = −1.26	0.21	0.22
**Effect of condition on perceptions of descriptive norms, injunctive norms, personal norms and expected satiety**					
Perceptions of descriptive norms ^a^	2.59 (1.03)	3.06 (1.09)	*t*(130) = −2.67	0.01	0.46
Perceptions of injunctive norms ^a^	2.68 (0.91)	3.18 (1.14)	*t*(130) = −2.66	0.01	0.46
Personal norms ^a^	2.99 (1.14)	3.39 (1.56)	*t*(130) = −1.19	0.24	0.21
Expected satiety ^a^	3.60 (1.36)	4.00 (1.63)	*t*(130) = −1.25	0.22	0.22

^a^ Measured by a 9-point scale (range 1–9). Note: All reported means and standard deviations are untransformed scores for ease of interpretation.

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
