# Peer review of "Served Portion Sizes Affect Later Food Intake Through Social Consumption Norms"

_nutrients, 2019, doi:10.3390/nu11122845_

Round 1
Reviewer 1 Report
Overall comment:
The authors present an interesting and well-conducted series of two experiments to investigate the effects of visual and actual exposure to small vs. large portion sizes on intended/actual food selection and consumption. I have a few comments/questions that I feel should be clarified and addressed.
Introduction:
The introduction is well-written and referenced. I especially appreciated the authors’ clear explanations of the various norms being investigated in the study (personal vs. descriptive social vs. injunctive social). Lines 90-93, hypothesis: I would have liked to have seen a more specific and directional hypothesis (similar to what was written in Lines 111-112) plus hypothesized mediators (i.e., social vs. personal norms) here.
Methods, Study 1:
Pilot study results are a significant strength that informed the main study design. Pilot study participants appear to be broadly representative of those in the main study (2/3 female and mid to late 20s). Measures: I appreciate that the authors included the specific text used to measure each outcome/mediator. I only wondered if this would be better formatted in a table vs. text (minor).
Results:
Table 1: Would ask the authors to confirm no differences in participant characteristics across conditions or note any differences if present (e.g., appears that gender distribution tended to vary). Lines 278-281, funneled manipulation check: I find it interesting that participants assigned to the smaller portion condition (in both studies) were much better at identifying the portion size to which they were visually exposed (Study 1) or served (Study 2). I would have expected to see more discussion of this finding in the conclusions for each study and/or the overall discussion. What might be driving this difference? Related to above, the exploratory analyses in both studies demonstrate a potentially stronger effect among those participants who were able to correctly identify the portion sizes to which they were exposed. Does this suggest a potential role of mindfulness training/mindful eating as an intervention strategy to reduce perceptions of “normal” portion sizes and reduce energy intake?
Methods, Study 2:
The authors note that gender was related to outcome and mediators in Study 1 so only females were recruited for Study 2. I have a few comments related to this decision. Did the effect of gender/sex persist when controlling for body weight and age? Would expect that portion sizes and their perceived normalcy would vary more so by body size (determinant of energy needs) than gender/sex. It appears as though the pilot study for Study 2 was conducted in a mixed-sex sample even though the main study was exclusive to females. This should be clearly stated and justified and/or noted as a potential limitation. The exclusion of men from the study also appears to be reflected in a general reduction in perceived portion size norms in Study 2 (Table 4) vs. Study 1 (Table 2). This leads me to wonder if there was something of a mismatch between the served portion sizes (based on the mixed-sex sample pilot) and the main study participants (all female). Line 406-409: I would just like to note that hunger and fullness are not necessarily the inverse of each other and I’m not sure it was entirely appropriate to combine a hunger question and reverse-coded fullness question as a measure of hunger. It does appear as though the Cronbach’s α is good though so this is likely only a minor issue. See: https://www.ncbi.nlm.nih.gov/pubmed/20122136 Line 433: change to 12pm Line 435: Were participants asked to abstain for exactly 2 hours vs. at least 2 hours vs. approximately 2 hours? Line 442: Was a 10g side salad served for both small and large portion size conditions? Line 446 and 461-462: Relates to comments I’ll make on results but it would nice to have data RE: serving/selection vs. consumption on both days as well as % of participants consuming entire serving on day 1.
Results, Study 2:
Similar comment as in Study 1 RE: manipulation check and ability of those assigned to the smaller portion to correctly identify served portion. Line 507-508 and Tables 4 and S7: I strongly believe the authors should present data on serving (Day 1)/selection (Day 2) vs. consumption on both days within the main manuscript rather than place some of these data in the supplementary content. I would suggest noting in the text that participants assigned to the smaller portion group consumed the entire serving (179.6 g consumed vs. 180 g served) while this was not the case in the larger portion group (450.5 g consumed vs. 540 g served). * I think this possibly relates back to the authors’ decision to use a mixed-sex pilot to inform a female-only study.* Also, I would suggest adding Selection (grams) from Table S7 into Table 4 along with Consumption (grams) under Effect of condition on consumption.Discussion:
Discussion is generally strong and integrates present research with the literature and suggests future areas of potential investigation. Would suggest addressing issues raised in this review in the discussion as appropriate.Author Response
Please see the attachment.

Reviewer 2 Report
I have reviewed the following manuscript “How do served portion sizes affect later food intake? The role of social consumption norms” and would recommend it for publication after some minor comments are addressed. The work is interesting, and important in understanding the role of norms in portion size selection and subsequent food intake. The manuscript is well-written (but would benefit from a couple of minor improvements). The 24-hour re-test and inclusion of a virtual and lab-based study strengthen the findings. The statistical methods are sound, the findings are not overstated, and the discussion is comprehensive and easy to read.
Lines 36-41
Recent insights show that portion sizes served in a given eating occasion not only affect immediate consumption, but also affect subsequent portion selection and consumption at later eating occasions [7,8]. Thus, changes to portion sizes in the environment have potential downstream consequences beyond a single eating occasion. However, relatively little is known about the mechanism responsible for this effect: why do people eat less (more) at later eating occasions when served smaller (larger) food portions?
It’s not clear where the reference is for the effect mentioned at the end of the first paragraph. I’m assuming it’s from the studies referenced immediately before (7,8). In addition, the last sentence seems a bit out of place. Reading this sentence gave me the impression that it was referring to an effect which was already introduced earlier in the paragraph, but that earlier sentence mentions the findings very generally – it does not provide specifics about how an eating occasion affects immediate and subsequent consumption (i.e., that people eat less (more) at later eating occasions when served smaller (larger) food portions). One solution would be to re-phrase the final sentence to “People eat less (more) at later eating occasions when served smaller (larger) food portions (REF). However, relatively little is known about the mechanism responsible for this effect.” This is all very minor, but I think addressing these comments might make that first paragraph clearer and flow a bit better.
Lines 61-64
Perceived ‘personal norms’ represent the amount of food consumers consider to be a normal amount for themselves to eat [10,11]. The construction of ‘personal norms’ (e.g., the belief about what is an appropriate amount for oneself to eat) is a dynamic process that is influenced by the external environment [10], and a served portion size may affect future consumption by affecting one’s ‘personal’ norm.
I find these two sentences to be confusing as each seems to define personal norms differently: “what is considered to be a normal amount for themselves to eat” vs “the belief about what is an appropriate amount for oneself to eat”. My understanding is that these are two different aspects of personal norms. In other words, what I normally eat might differ from what I think is an appropriate amount to eat for myself. I might normally eat a 700 kcal meal (because that’s the amount that leaves me feeling satisfied), but believe that I should be eating a 500 kcal meal because of my current weight status or activity levels. This distinction in personal norms seems to be made in the methods section (based on the two “personal norms” questions: ‘how much of this [lasagne/spaghetti] do you personally believe is a normal amount for you to eat for lunch?’ VS ‘how much of this [lasagne/spaghetti] do you personally believe is an appropriate amount for you to eat for lunch? However, these two sentences in the introduction need to be reworked to more clearly define and distinguish these two facets of personal norms. Are these sentences trying to communicate that a personal norm is defined as “what is considered to be a normal amount for the individual”, and that the way these personal norms develop is through beliefs about one’s needs (i.e., how much food will leave them satisfied, and how much food is necessary for their gender, body size, activity levels)? If so, I think that needs to be made clearer.
Lines 183-186
Why was it decided to measure hunger retrospectively? Is there evidence that individuals are accurate in recalling hunger/fullness states?
Lines 221-223
I’m assuming that the color intensity of the images changed across the 14 dimensions? This could be clearer.
Line 231
It’s misleading to say participants reported their “current hunger” as Lines 183-186 explain that hunger was rated retrospectively.
Line 424-429
Exclusion Criteria
Awareness of the aims of the study was assessed using the same methodology as in Study 1. Agreement for coding decisions between two independent researchers was 100%.
Instruction following was measured with the following item: ‘how long ago did you last eat?’ (options: 30 minutes ago, 1 hour ago, 1.5 hour ago, 2 hours ago, 2.5 hours ago, 3 hours ago and longer than 3 hours ago).
I know this is mentioned under “planned statistical analyses” but I think it might be helpful to clarify here that these are not exclusion criteria for taking part in the study, but were exclusion criteria for analysis. Same with Study 1. I also think “Instruction following” is vague and could be made more specific. Something like “adherence to abstinence requirements” might be better?
Line 496-498
As in Study 1, Pearson chi-square analysis showed that a higher proportion of participants in the smaller portion size condition (68/68, 100%) were able to identify the portion size they consumed in session one than in the larger portion size condition (24/64, 37.5%), X2(1) = 60.98, p < .001. Results of a Welch’s t-test indicated that participants in the smaller portion size condition (M = 1.32, SD = .47) remembered being served a significantly smaller portion than participants in the larger portion size condition (M = 5.94, SD = 1.83) (t(70.82) = -19.54, p < .001, d = 3.51), however the proportions were higher than in Study 1, suggesting better recall of portions when they are served and consumed than when visually presented online
I see that the analyses were re-run using a subsample (removing those who incorrectly recalled the portion size from session 1) and find somewhat different results (warning that they should be interpreted with caution due to uneven sample sizes), but I don’t see any discussion of why there might be better recall for portion size in the smaller portion size condition (compared to the larger portion size condition)? I think this is interesting and should be briefly mentioned in the discussion.
Line 595
There’s a typo – “visual” should say “visually”
Round 2
Reviewer 1 Report
Thank you for thoroughly addressing my comments from the initial review.
Author Response
Thank you for reviewing our manuscript.